# Innovative Inorganic Binder Systems for the Production of Cores for Non-Ferrous Metal Alloys Reflecting the Product Quality Requirements

Václav Merta [1,*], Jaroslav Beňo [1], Tomáš Obzina [1], Filip Radkovský [1], Ivana Kroupová [1], Petr Lichý [1], Martin Folta [2], Kamila Janovská [1], Isabel Nguyenová [3] and Miroslav Dostál [3]

[1]  Faculty of Materials Science and Technology, VSB—Technical University of Ostrava, 708 00 Ostrava, Czech Republic; jaroslav.beno@vsb.cz (J.B.); tomas.obzina@vsb.cz (T.O.); filip.radkovsky@vsb.cz (F.R.); ivana.kroupova@vsb.cz (I.K.); petr.lichy@vsb.cz (P.L.); kamila.janovska@vsb.cz (K.J.)

[2]  Department of Production, Logistics and Quality, ŠKODA AUTO University, 293 01 Mladá Boleslav, Czech Republic; martin.folta@savs.cz

[3]  Brembo Czech, s.r.o., 720 00 Ostrava, Czech Republic; Isabel_Nguyenova@cz.brembo.com (I.N.); Miroslav_Dostal@cz.brembo.com (M.D.)

*  Correspondence: vaclav.merta@vsb.cz; Tel.: +420-608-863-842

**Abstract:** The aim of this study is the evaluation of the parameters of core mixtures using different binder systems with regard to the collapsibility of cores after casting and the resulting product quality of castings reflecting surface requirements based on non-ferrous alloys. The research compares organically bonded core mixtures based on phenol-formaldehyde resins for the production of cores with the shell molding (resin coated sand), currently used in the production of aluminum alloy castings in the Brembo Czech s.r.o., and mixtures using innovative inorganic binder systems based on geopolymers; GEOPOL® W. The aim of the research is to compare the advantages and disadvantages of these binder systems in order to evaluate the potential of inorganically bonded mixtures to replace organically bonded mixtures, which would lead to a significant reduction in the environmental impacts of industrial production of castings.

**Keywords:** foundry; aluminum casting; core; inorganic binder; collapsibility; surface product quality

## 1. Introduction

Although inorganic binders have been well known in foundry technology since 1947 according to the Czechoslovak patent [1], their use for the production of molds and cores is still not very widespread. Although they are environmentally friendly, as they are the source of water vapor or carbon oxides only during casting, solidification and cooling of castings, their widespread application is limited mainly by lower mechanical properties in comparison with traditional organic binder systems [2–4]. Other negative factors include their fragility, lower productivity of the core manufacturing process, limited storage life of the produced cores (e.g., maximum up to 2 weeks for sodium silicate cured with $CO_2$), increasing of technical requirements for the quality of the final casting products and especially poor collapsibility of the cores after casting. This is caused by the phenomenon when at temperatures above 793 °C the strength increases to the area of the so-called II. strength maximum, and the resulting strengths are many times higher than the original (primary) strength after curing. The reason is the formation of a glass melt of the binary system $SiO_2$-$Na_2O$, which also melts the surface of the silica sand and at high temperatures the interface binder film–grain surface disappears, so a compact monolith with high residual strength is formed. The main component supporting the formation of the melt in the mixture is $Na_2O$, which is the basis of the most common inorganic foundry binders—sodium silicate ($Na_2O \cdot SiO_2 \cdot H_2O$) [5].

The break point of inorganic binder application was 2003, when the traditional Foundry Fair GIFA (International Foundry Fair GIFA 2003 in Dusseldorf) was held, where innovative inorganic binder systems were introduced. Their common characteristic is a different way of curing by dehydration (reversible) procedures according to Equation (1):

$$Na_2O \cdot mSiO_2 \cdot H_2O + energy \underset{+H_2O}{\overset{-H_2O}{\underset{\leftarrow}{\rightarrow}}} Na_2O \cdot mSiO_2 \tag{1}$$

$Na_2O \cdot mSiO_2 \cdot nH_2O$—sodium silicate hydrate.
$Na_2O \cdot mSiO_2$—sodium silicate.

Core mixtures with inorganic binders cured by dehydration processes achieve comparable or higher strengths in comparison with the mechanical properties of core mixtures with traditional organic binders [6,7]. The binder content could therefore be reduced from the original dosage of 4–5% binder (sodium silicate cured with $CO_2$) or 2–3% binder (ester cured self-setting sodium silicate process) to 1.6–2.2% for innovative inorganic binders. In addition to the economic benefits, this reduced dosage has a positive effect on the reclamation of these molding mixtures and improving their collapsibility. The productivity of core manufacturing process is also increased and it is comparable to traditional organic binders. The rate of the curing process is mainly influenced by the speed of water vapor removing from the system. Therefore, there are different ways of curing innovated inorganic binders (cold or hot air blowing, warm or hot box, microwave curing, etc.) or their combination. The limiting factor of most innovative inorganic binders is the storage life of the produced cores. Since the principle of curing is a dehydration process (reversible process), in the case of increased relative humidity of the ambient atmosphere, the cores absorb the humidity and thus degrade. In addition, the use of water-based coatings, which are more environmentally friendly than alcohol-based coatings (especially for the working environment), is limited. Conversely, the reversibility of the curing process can be used for simplified knocking—out or improvement of core collapsibility. For this reason, the use of innovative inorganic binders is found mainly in the foundries of large car manufacturers (ŠKODA AUTO a.s., VW, BMW, etc.). The production in the automotive industry has been realized under very strict conditions, with high demands on the product quality of parts manufactured by car makers and components delivered by their suppliers, which result from the ever-increasing product quality requirements of final car makers [6–10].

As the production of automotive components represents the production of very complex castings, especially from non-ferrous metal alloys, mainly aluminum alloys, the most widespread core production technologies include COLD-BOX Amin or resin coated sand (RCS). Although these binders have undergone a number of modifications in an effort to reduce the environmental footprint (solvent change, application of inorganic binder additives, reduction of free formaldehyde and phenol), they are still a source of many hazardous substances (VOCs, PAH, etc.) during casting production. In addition, although lower thermal decomposition binders are used for the production of castings from low heat capacity alloys (e.g., silumins), in some cases satisfactory core collapsibility is not achieved. This is due to the fact that the temperature reached in the cores is lower than the destruction temperature of the organic resin. This phenomenon is especially evident in the production of thin-walled and complex cores. The removal of cores is then most often solved by repeated annealing of castings to temperatures higher than the temperature of thermal destruction of the organic binder (usually 400 °C for castings from aluminum alloys). This additional process is a source of increased production costs and reduced productivity, especially for castings that are not further heat treated in order to increase the mechanical and utility properties of cast structural components. There is also a potential risk that the quality requirements of the final product will not be met, and thus increase the total cost of poor quality.

The aim of the presented paper is to verify the possibility of application of the innovative inorganic binder system GEOPOL® W in comparison with the traditional core

manufacturing technology applied in the production of automotive castings—RCS technology. At first, the mechanical properties of individual thermosetting binder systems in the form of hot and cold strengths were evaluated. The comparison of the collapsibility rate was performed on test castings using a modified methodology for evaluating the collapsibility of traditional inorganic binders (CO$_2$—sodium silicate, ester cured self-setting sodium silicate process) developed at VSB—Technical University of Ostrava. The applicability of the selected binder system was further evaluated with respect to the surface quality of test castings.

## 2. Materials and Methods

The basic principle of shell molding, generally referred to as resin coated sand (RCS), is the melting of a thermoplastic phenol-formaldehyde resin of the NOVOLAK type, which is formed by polycondensation of phenol and formaldehyde in an acidic environment (in a ratio of 1:0.4–1:0.9) in the presence of mineral acids or strong organic acids as catalysts [5].

The shell process gets its name from the ability to make either solid or hollow cores with a thin wall or shell, where the unused sand is removed from the core cavity or mold surface and reused. Shell cores have always been associated with the process of high-quality casting production but concerns with emissions, smell and productivity. Recent advancements in resin-coated sand technology have addressed many of the concerns with emissions and productivity. Shell users have typically been concerned with free phenol, free formaldehyde and ammonia levels in their facility. By changing the chemical makeup of the reactants used to form and cure shell sand, suppliers now offer a resin-coated shell sand with fewer hazardous air pollutants and reduced odor [11].

GEOPOL$^®$ is a unique inorganic binder system and this technology is currently used in the foundries for three basic production processes/technologies: (1) for self-hardening molding mixtures, (2) sand mixtures hardened by gaseous carbon dioxide, and (3) the hot box technology with hot air hardening. The geopolymers with a high molar ratio of SiO$_2$/Al$_2$O$_3$, sometimes called geopolymer resins, are liquid substances with similar properties to colloidal solutions of alkali silicates—sodium silicate.

These polymers are also referred to as polysialates and are composed of chains of tetrahedrons of SiO$_4$ and AlO$_4$ [12]. The resulting properties of the binder depend on the ratio of these components and on the preparation of the geopolymer [13].

The binder is an inorganic geopolymer precursor with a low degree of polymerization. The hardening occurs by the action of heat or hardeners. There is an increase in the degree of polymerization and formation of an inorganic polymer during the hardening reaction.

The GEOPOL$^®$ is odorless technology and generates no pollutants, so it has a minimal negative impact on the environment (Figure 1).

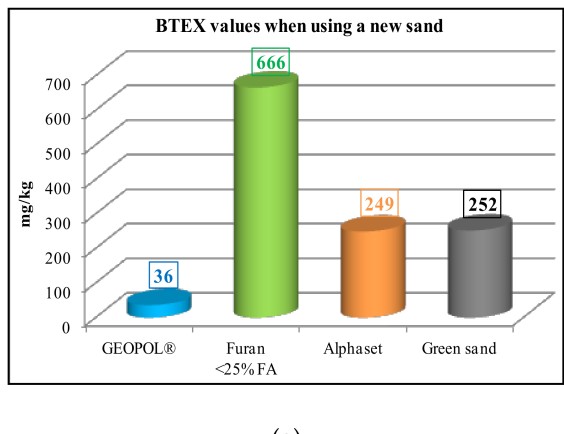

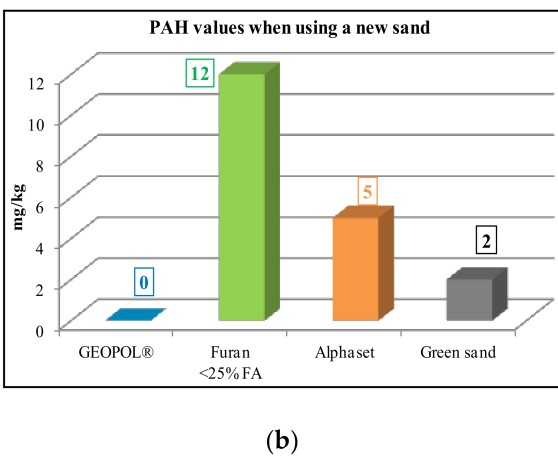

(**a**)　　　　　　　　　　　　　　　　　(**b**)

**Figure 1.** Comparison of ecological performance of organic and inorganic binder system during pouring: (**a**) BTEX values and (**b**) PAH values [13].

Following general parameters commonly used for characterization of core mixtures were determined:

1. Loss on ignition of dried samples (105 °C up to constant weight) at 900 °C/2 h,
2. pH and conductivity of water suspension (1:10 solid–liquid ratio),
3. Particle size distribution according to German standard VDG MERKBLATT P34.

Test specimens measuring 22.4 mm × 22.4 mm × 170 mm for determining the mechanical properties of mixtures with the studied binders were prepared on a laboratory shooting machine LUT, Multiserw Morek in a triple corebox. The production parameters of test specimens for the determination of transverse strength and for the production of test cores are summarized in Table 1. The setting of process parameters for the production of RCS cores is based on the standard operating conditions used in Brembo Czech. For mixtures with geopolymer binders, the parameters recommended by the binder supplier, SAND TEAM, spol. s.r.o.

**Table 1.** Parameters for the production of test specimens for the evaluation of mechanical properties of mixtures and test cores.

| Parameters of Curing | Binder System | |
| --- | --- | --- |
| | GEOPOL® W | RCS |
| Corebox temperature | 150 °C | 280 °C |
| Shooting pressure | 6 bar | 2 bar |
| Shooting Time | 6 s | 2 s |
| Curing Time | 120 s | 120 s |
| Amount of mixture in shooting head | 700 g | 600 g |

The middle core was used to determine hot transverse strength (measured immediately after preparation), the outer cores were used to determine cold transverse strength after 1 h from preparation. Transverse strengths were measured using universal testing machine Multiserw Morek, the LRU-2e type. Samples of individual studied RCS were used directly for the production of test specimens on the LUT device. The samples of the core mixtures with inorganic binder GEOPOL® W were prepared by 1 min homogenization of the silica sand with powder additive, W303 or W308B, in the constant weight ratio 1:100 and then the constant amount 2% of the binder GEOPOL W20 were added to the mixture for further homogenization for another 1 min.

Mixtures of the same composition as for the evaluation of mechanical properties were also used to evaluate the collapsibility of core mixtures with the studied binders. For the purpose of evaluating the collapsibility of core mixtures, a new test method based on the measurement of the impact energy required to break through the core stuck in the test casting using a laboratory sand rammer with a special thorn device was proposed (Figure 2), which was derived from the test methodology of the Technical University of Ostrava for the evaluation of the collapsibility of molding mixtures with sodium silicates [5].

The test casting (Figure 3) weighing 196 g (gross weight 401 g) with an average wall thickness of 5.0 mm and dimensions of 32.5 mm × 65 mm × 60 mm was designed to simulate the production of thin-walled castings with relatively massive cores, in which insufficient core heating occurs, and thus the thermal destruction of core mixtures with organic binders is considerably limited. The low heating of the cores is then the reason for the deteriorated cleanability of the castings, respectively impaired collapsibility of mixtures with organic binders. In this study, the low thermal stress of the cores was ensured by a low core/cast weight ratio of about 1:2. In addition, it is necessary to take into account the change of stress states at the core/metal interface and the increase of the internal stress in the core, which results in the "clamping" of the core by the shrinking casting.

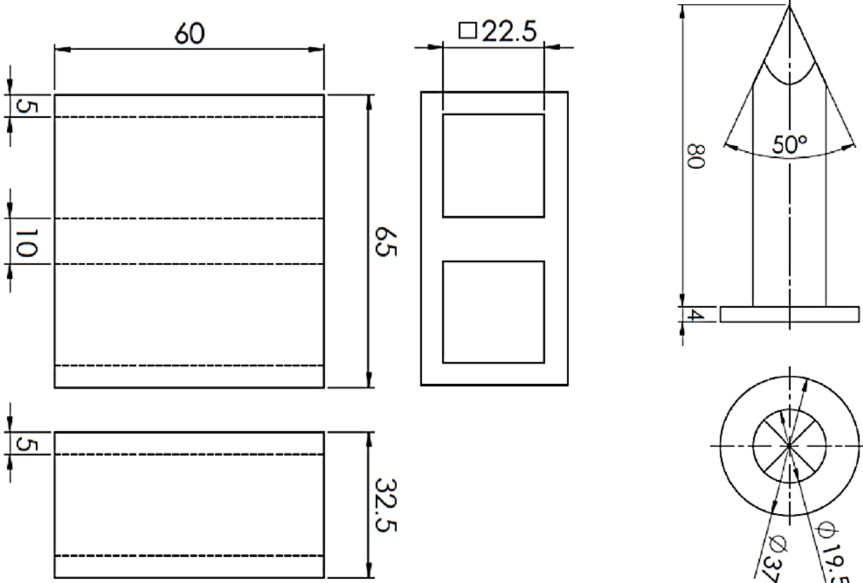

**Figure 2.** Scheme of collapsibility test method (mm).

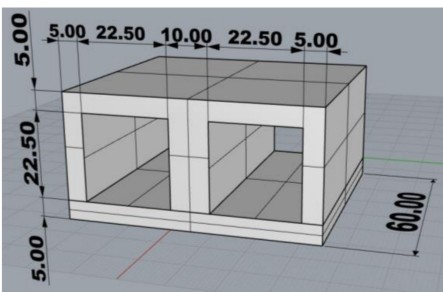

**Figure 3.** Drawing of the test casting (mm) (3D, Rhinoceros 6).

An ester cured self-setting sodium silicate process was used to make the molds. Due to its properties (especially with regard to heat dissipation), this system is very close to the effect of metal molds (accords to real operating conditions) [14]. These assumptions were verified before the actual casting tests using the simulation program MAGMASoft. The aim of the simulation was mainly to verify the functionality of the proposed casting technology and to predict the heating of the tested cores. Input parameters of the intended production technology were used to set the simulation (casting—aluminum alloy AlSi7Mg, mold—ester cured self-setting sodium silicate process, cores—RCS and casting temperature 720 °C). Figure 4 shows the geometry of the test casting including the inlet system, vents and cores. Figure 5 shows the prediction of temperature distribution in the cores immediately after pouring.

From the performed simulation it is evident that the achieved temperature of the test cores after casting is significantly lower (25–229.5 °C) than the temperature of thermal destruction of the Novolak resin used for RCS technology (approximately 600 °C).

The test castings were cast from AlSi7Mg aluminum alloy with a constant casting temperature of 720 ± 5 °C and with a constant casting time of 4 ± 0.5 s.

The surface quality of test castings around cores with studied binder systems was determined using a digital microscope KEYENCE, HX 6000. The surface quality of test castings was compared visually, in the form of photographs (overall view at 10× magnification and surface detail at 500× magnification in a high dynamic range—HDR), and also using the measurement of the mean arithmetic roughness Ra and the maximum peak to valley height of profile Rz determined from ten points in accordance with the Czech

standard ČSN EN ISO 4287. The measurement was performed on the upper half of the test casting. In the area around the core, five measuring points with a length of 4000 μm were chosen so that the surface quality was determined over the entire studied area. The range $0.6 < Ra < 2.0$ μm was chosen for the evaluation of parameters Ra and Rz, $\lambda_c$ was chosen 0.8 mm and $\lambda_s$ 25 μm in the first in row mode.

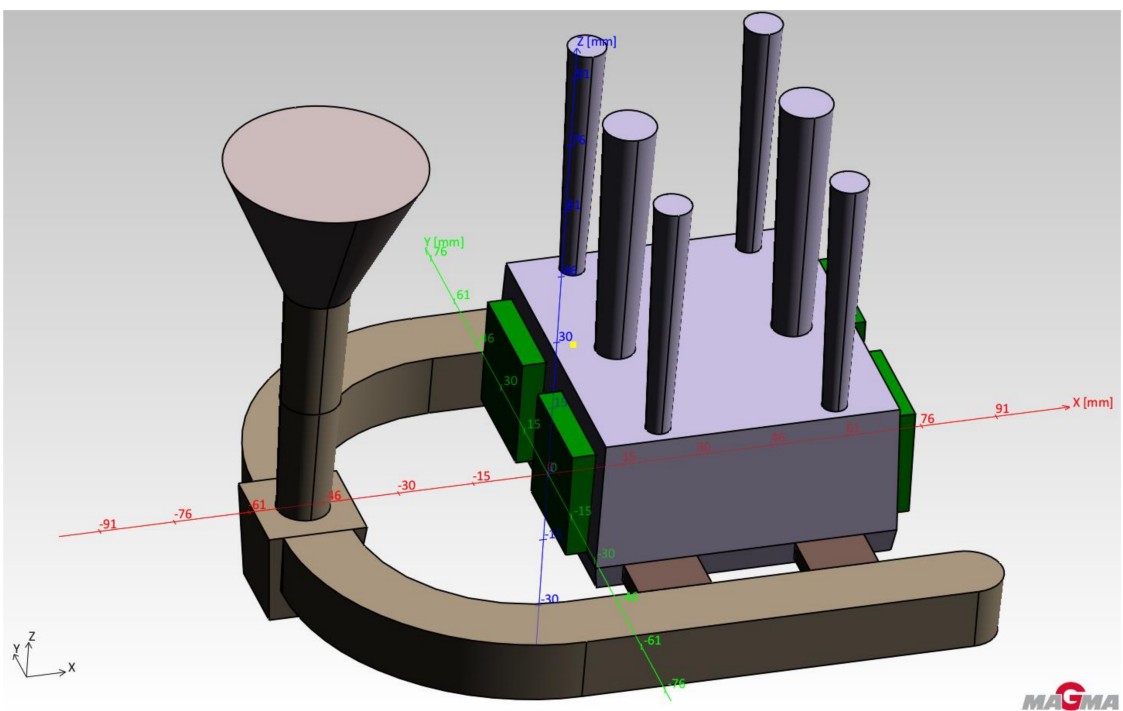

**Figure 4.** Scheme of testing casting.

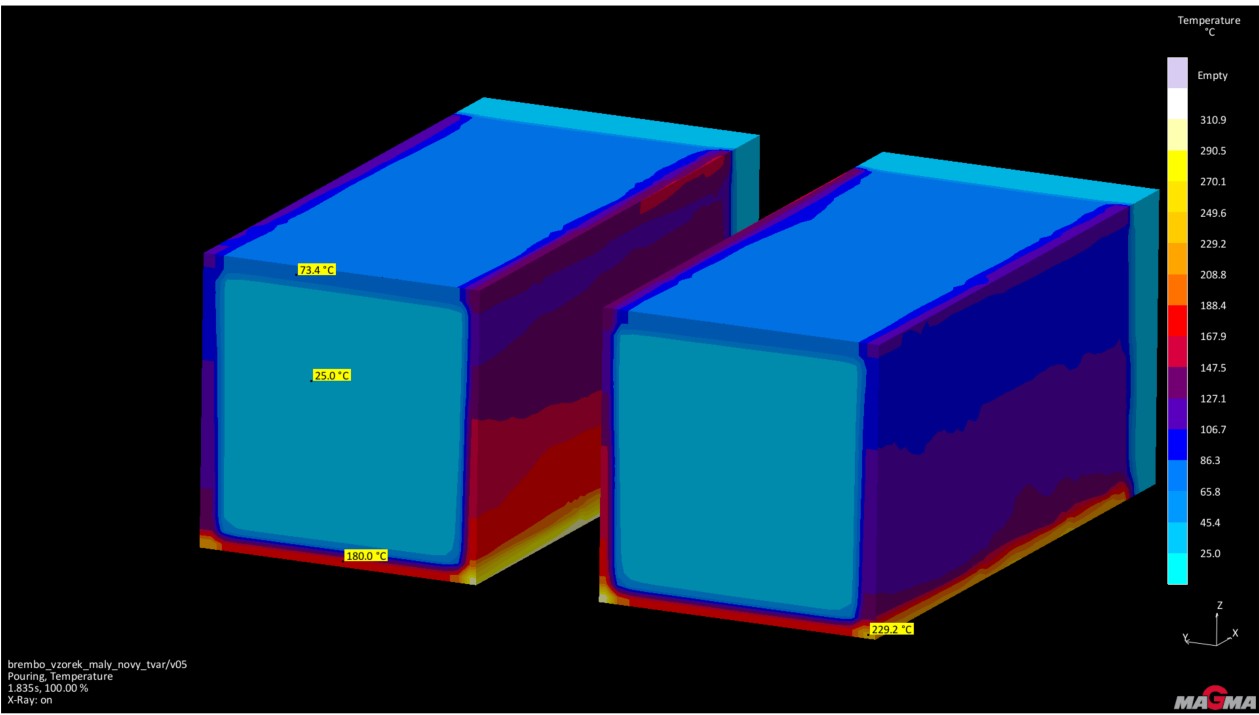

**Figure 5.** Temperatures of testing cores after pouring.

### 3. Results

Four different core systems were used to verify the possibility of using core mixtures with innovative inorganic binders: (I) GEOPOL® W20 binder with GEOTEK W303 powder additive (marked as W303); (II) GEOPOL® W20 binder with GEOTEK W308B powder additive (marked asW308B). Two commonly used RCS core mixtures used in the conditions of Brembo Czech were selected as a standard: (III) RCS CB20B (marked as CB20B) and (IV) RCS DA30 (marked as DA30). The basic parameters of the test mixtures are summarized in Table 2.

**Table 2.** General parameters of studied core mixtures.

| Parameter | Unit | W303 | W308B | CB20B | DA30 |
|---|---|---|---|---|---|
| Loss on ignition | % | 0.28 | 0.34 | 1.79 | 2.52 |
| pH | - | 10.71 | 10.76 | 6.93 | 8.01 |
| Conductivity | μS/cm | 1159.0 | 908.0 | 36.2 | 26.7 |
| Medium grain size $d_{50}$ | mm | 0.24 | 0.24 | 0.20 | 0.15 |
| AFS. GN. | - | 57.77 | 57.77 | 69.49 | 86.00 |
| Degree of homogeneity S [1] | % | 62.13 | 62.13 | 68.03 | 73.73 |
| Amount of fine particle (below 0.18 mm) | % | 19.55 | 19.55 | 37.23 | 80.44 |

[1] Calculated as the $d_{75}/d_{25}$ ratio.

The value of loss on ignition in mixtures with organic binders indirectly defines the content of binder in the mixture (amount of heat-degradable substances). The highest value is reached for the RCS system DA30 (2.52%), the lowest values were found for the binder CB20B (1.79%). For mixtures with inorganic binders, the difference in the loss on ignition values (0.28% for W303 and 0.34% for W308B) is only due to the nature of the powder additive, as the amount of GEOPOL® W binder dosed into the mixture is constant (2%). Low values of loss on ignition are due to the nature of the binder, which is not decomposable by heat (aluminosilicates). Dried samples are used to determine the loss on ignition, so the water content of the binder does not matter.

The values of pH and electrical conductivity confirm the diametrically different nature of the individual binders studied and the different structural composition of the binders. Mixtures based on GEOPOL® W binder confirm a strongly alkaline system, where the pH value ranged from 10.71 (W303) to 10.76 (W308B). From the achieved results it is clear that the powder additive did not have a significant effect on the overall pH value of the system. On the other hand, the group of binders belonging to the RCS systems showed a lower pH, the value was found to be 6.93 for the CB20B system and 8.01 for the DA30. This difference is probably due to the different resin used in the RCS system.

The mutual difference of the studied binders was also confirmed by a significant difference in the values of electrical conductivity, which ranged from 908 (W303) to 1159 μS/cm (W308B) for inorganic systems. These high values of electrical conductivity are due to the inherent structure of inorganic binders based on geopolymers. On the contrary, the minimum values of electrical conductivity were found in RCS systems. The conductivity ranged from 26.7 for CB20B to 36.2 μS/cm. These low values only confirm the different nature of the two groups of binders (RCS has a minimum of free cations).

Significant differences were found in mixtures with the studied binders in terms of sieve analysis. The mean grain size $d_{50}$ ranges from 0.15 mm (DA30), through to 0.20 mm (CB20B) to 0.24 mm for both GEOPOL® inorganic binder systems. The value of AFS GN correlates with this (for W303 and W308B AFS GN = 57.77; for CB20B AFS GN = 69.49 and finally for DA30 AFS GN = 86.00). This is in accordance with the well-known rule that the value of AFS GN increases with decreasing value of the mean grain $d_{50}$. The different fineness of the sands used in the individual mixtures is also confirmed by the proportion of fine particles below 0.18 mm, which ranges from 19.55% for mixtures W303 and W308B, through 37.23% (CB20B) to 80.44% for the DA30 system.

On the one hand, differences in sieve analysis can affect the mechanical properties of the mixture. As is generally known, the high proportion of fine particles must be compensated by an increased dosage of binder (as the size of the individual particles decreases, the total surface area of the sand increases significantly, thus increasing the consumption of binder to achieve the required strengths). However, this could have a negative effect on the gas regime of the mold (core)–metal system and the tendency to gas defects formation [15]. On the other hand, a required surface quality of the casting is generally ensured and the potential for metal penetration types of casting defects was significantly eliminated. For systems with an inorganic binder, i.e., W303 and W308B, sand with a particle size distribution as is commonly used in the production of aluminum alloy castings (AFS GN 57.77) was chosen. It is a sand with individual fractions distributed over several sieves, which confirms the calculated degree of homogeneity S. This is defined as the proportion of $d_{75}/d_{25}$ found from sieve analysis. The values of the degree of homogeneity (S) range from 0 up to 1, and when the value of S is closer to 1, the more monofractive sand it is, and vice versa. The value of S therefore tells us whether it is made up of several fractions or has proportions on only a few sieves. This is generally used, since a suitable composition of the individual sand fractions makes it possible to ensure maximum strength of the mixture with maximum permeability of the mixture using the given sand. It follows from the above that the DA30 and CB20B systems show a steep sum curve, as the degree of homogeneity S takes values of 68.03% (CB20B) and 73.73% (DA30). In contrast, the sand used for the W303 and W308B systems has a flatter sum curve (S = 62.13%).

### 3.1. Mechanical Properties of Mixtures with Studied Binders

One of the key parameters influencing not only the surface quality of precast holes and internal cavities of cast components, but also a number of other quality parameters, are the mechanical properties of cores under normal conditions (primary strength). However, the possible occurrence of a number of foundry defects is also determined by hot strength. An overview of the mechanical properties of core mixtures with the studied binders are summarized in Table 3. The stated hot transverse strength is the average of three measurements, the cold transverse strength is the average of six measurements.

**Table 3.** Mechanical properties of mixtures with tested binders.

| Mixture | Hot Transverse Strength | | | |
|---|---|---|---|---|
| | Sample Weight (g) | Sx (g) | Strength (Mpa) | Sx (MPa) |
| W303 | 133.36 | 0.25 | 4.80 | 0.16 |
| W308B | 131.48 | 0.24 | 3.68 | 0.03 |
| CB20B | 133.71 | 1.27 | 2.21 | 0.12 |
| DA30 | 127.37 | 2.15 | 2.95 | 0.40 |
| **Mixture** | **Cold Transverse Strength** | | | |
| | Sample Weight (g) | Sx (g) | Strength (Mpa) | Sx (MPa) |
| W303 | 131.52 | 0.88 | 7.37 | 0.27 |
| W308B | 130.49 | 1.39 | 7.33 | 0.27 |
| CB20B | 131.83 | 2.63 | 4.47 | 0.41 |
| DA30 | 126.16 | 2.77 | 5.10 | 0.91 |

The achieved results show that the highest hot transverse strength was achieved with mixtures using the inorganic binder GEOPOL® W, 4.80 MPa for the W303 system and 3.68 MPa for the W308B system. The same trend was observed for cold transverse strength. Again, the highest transverse strengths were found for the W303 system (7.36 MPa), however, the same strengths were achieved for the W308B system (7.33 MPa).

The difference in cold strengths between these mixtures was thus only 0.46%. For clarity, the achieved transverse strengths were expressed in relative values (Figure 6).

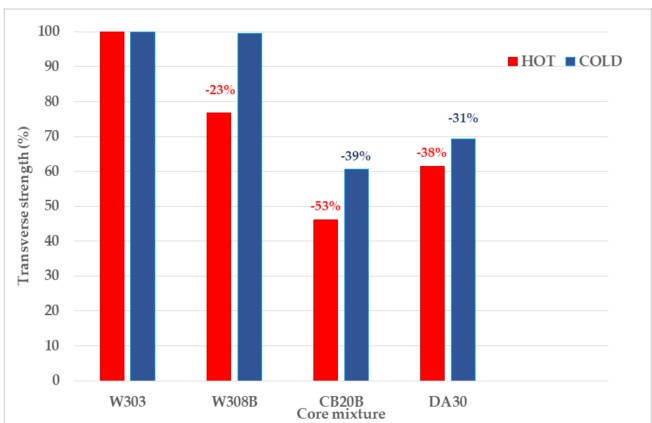

**Figure 6.** Comparison of relative values of cold and hot transverse strengths of tested mixtures (relative to the mixture with the highest strengths—W303).

The maximum difference transverse strength between individual RCS and mixtures with inorganic binders was up to 53% for hot strength and 39% for cold strength. This difference could be caused by different injection pressures in the production of test specimens (2 bar for RCS, 6bar for inorganic binders), however the weight of the test specimens ranges from 127.37 (DA30) to 133.36 g (W303). Thus, this difference was not so significant that the difference in achieved strengths was 39%.

Such high mechanical properties of molding mixtures are due to the high mobilization of the inorganic binder cured by dehydration processes in comparison with the usual chemical way (sodium silicate cured with $CO_2$, ester cured self-setting sodium silicate process). This effect is also the reason why inorganic systems can replace the most common core systems (COLD—BOX Amin) in the production of castings. It follows from the above that it would be possible to reduce the dosage of the inorganic binder in the test mixture. This intervention would not only have an economic effect, but would also improve the flowability of the mixture, it would be possible to produce test specimens with lower injection pressures while maintaining the required quality of the cores. In general, a lower binder content improves not only the reclaimability of the system, but also the collapsibility of the mixture (cores knocking—out).

The lower binder content of the mixture also has a positive effect on the gas evolution from the cores. One of the advantages of RCS compounds is the ability to produce hollow cores. Among other things, gases that are formed during the decomposition of the organic binder are removed by the cavity in the core. In general, only solid cores can be produced from mixtures with the inorganic binder Geopol®-W, however, the amount of released gases, as with other inorganic binders [16–18], was significantly lower (Figure 7) compared to conventional organic binder systems, as already reported in previously published studies [19].

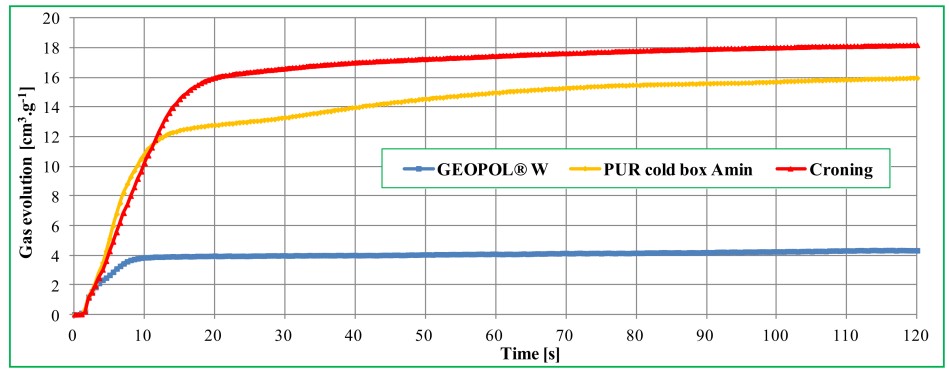

**Figure 7.** Comparison of gas evolution values of different core mixtures [19].

### 3.2. Influence of Binder Type on the Collapsibility of the Core Mixture

The collapsibility of molding or core mixtures is one of the most important characteristics of a binder system. Very often it is a decisive factor in the question of the suitability of its use for a given application. As mentioned above, the collapsibility of the mixture, which can also be evaluated as residual (secondary) strength, is directly correlated with the primary strengths. Particularly in the case of inorganic binders, which are not subject to thermal destruction due to the heat released from the solidifying and cooling casting, the higher the primary strength (after curing), the higher the residual strength and the worse collapsibility. A comparison of the collapsibility of mixtures with the studied binder systems is summarized in Figure 8.

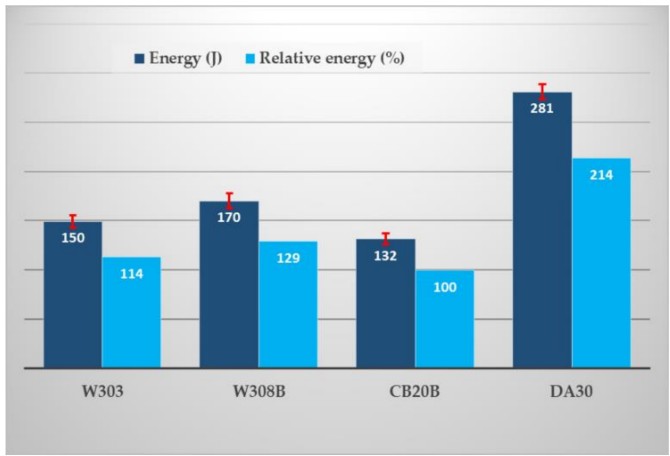

**Figure 8.** The collapsibility of studied binder systems.

The collapsibility of the mixture was defined as the impact energy required to break through the core stuck in the test casting. This value is the result of an evaluation on six cores (three castings), excluding outliers.

From the obtained results it is evident that the resulting collapsibility of the mixtures with the studied inorganic binders is comparable or better in comparison with the RCS mixtures. This is probably due to the embrittlement of the binder gel during thermal exposure from the cooling casting. The effect of thermal expansion in the case of the use of silica sand can also play a role, but only under the assumption of higher thermal exposure of the core (reaching a temperature of approximately 573 °C and the associated modification changes alpha-quartz—beta quartz).

Other factors also play a role: (i) the type of powder additive for inorganic systems. With the same binder content (2%), the impact energy ranges from 150 (W303) to 170 J (W308B), an increase of 12%. (ii) The effect of the binder content as such is also apparent. The RCS mixture CB20B mixture achieved an impact energy of 132 J, with a binder content of 1.8% (see Table 2). On the other hand, the value of impact energy 281 J in the DA30 system is probably due to the highest binder content (2.8%) of all studied mixtures. Related to this is the possibility of influencing the collapsibility of mixtures with inorganic binders. With regard to the achieved mechanical properties, both hot and cold, there is a possibility to reduce the impact energy by lower binder dosage (reduction from 2.0% to 1.8%), which is in accordance with the general rules for controlling the mixture with inorganic binders.

### 3.3. Surface Quality of Test Castings

The surface quality of test castings around cores from mixtures with studied binders was evaluated visually in the form of surface photographs at 10× and 500× magnification and using mean arithmetic roughness Ra and the maximum peak to valley height of profile Rz determined from ten points in accordance with the Czech standard ČSN EN ISO 4287. The upper half of the casting was always evaluated. The reason was the effort to evaluate

not only potential surface defects from the core, but also gas defects, which generally appear in the upper parts of the castings. The actual measured values of surface roughness of test castings are summarized in Table 4. The stated values of Ra and Rz are expressed as the average value from five measurements over the entire monitored area of the test casting.

**Table 4.** Surface roughness of test castings.

| Parameter/Mixture | Unit | W303 | Sx (µm) | W308B | Sx (µm) | CB20B | Sx (µm) | DA30 | Sx (µm) |
|---|---|---|---|---|---|---|---|---|---|
| Ra | µm | 5.44 | 1.03 | 3.28 | 0.25 | 2.48 | 0.17 | 4.77 | 0.41 |
| Rz | µm | 23.34 | 2.29 | 16.64 | 1.12 | 12.77 | 3.00 | 21.62 | 2.20 |

The achieved results show a significant difference in surface quality around the cores of test castings. The effect of different particle size distribution of used sands (AFS GN or medium grain $d_{50}$ values) is evident. On the surface of the casting around the cores of mixtures with mixtures with the inorganic binder GEOPOL® W (W303 and W308B), traces of sand grains imprinted on the surface of the casting are visible in the images at a magnification of 500×. This is due to the fact that sand with AFS GN of about 58 was used, while for the RCS system CB20B AFS GN was about 70 and for DA30 it was 86. However, it is a raw cast surface that reaching the quality requirements.

The surface quality of the test castings around the cores is high, the value of the mean arithmetic roughness Ra ranges from 2.48 (CB20B) to 5.44 µm (W303). In general, the values of Ra and Rz are in the area of surface roughness for castings produced by the HPDC method [20]. The surface quality of the test castings is summarized in Figures 9–16.

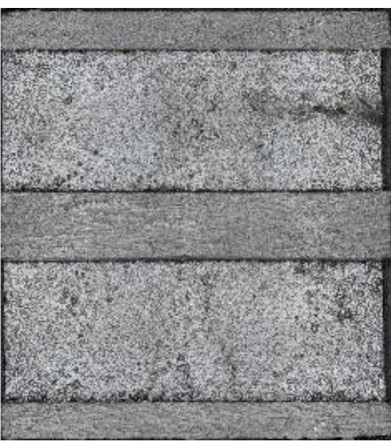

**Figure 9.** Surface quality W303, magnification 10×.

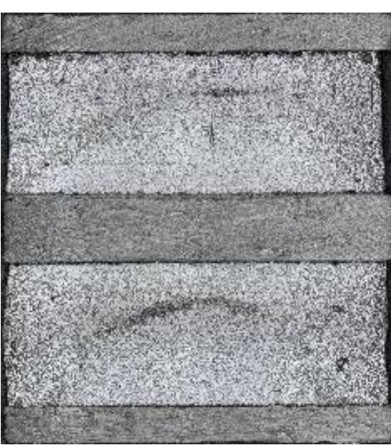

**Figure 10.** Surface quality W308B, magnification 10×.

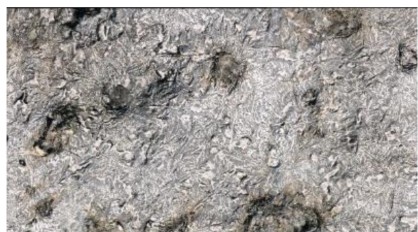

**Figure 11.** Surface quality W303, magnification 500×.

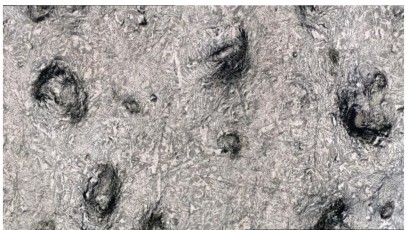

**Figure 12.** Surface quality W308B, magnification 500×.

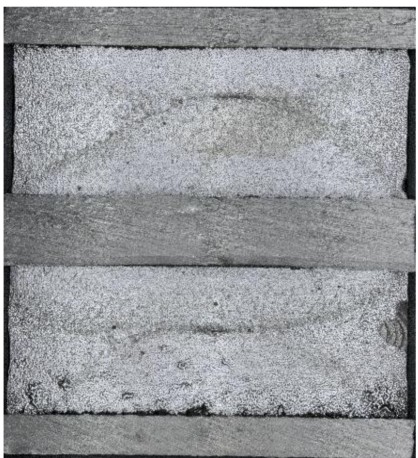

**Figure 13.** Surface quality CB20B, magnification 10×.

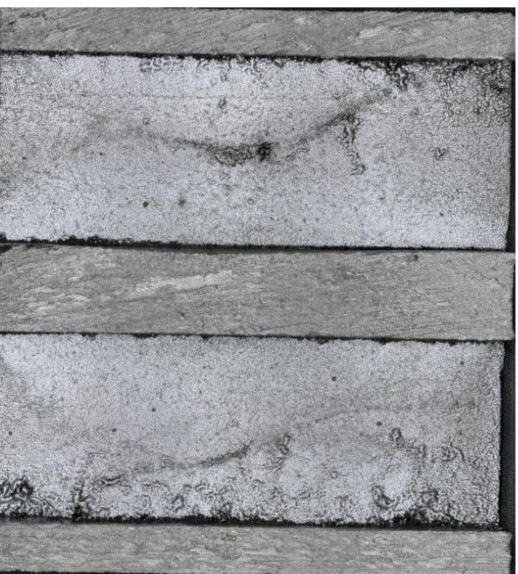

**Figure 14.** Surface qualityDA30, magnification 10×.

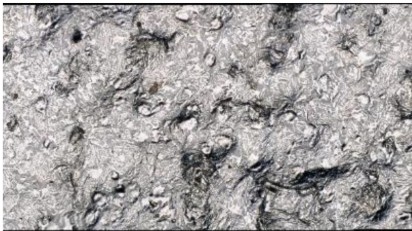

**Figure 15.** Surface quality CB20B, magnification 500×.

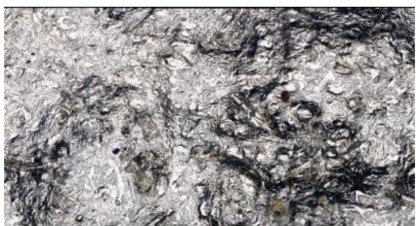

**Figure 16.** Surface quality DA30, magnification 500×.

## 4. Conclusions

The aim of the presented paper was to verify the possibility of application of the innovative inorganic binder system GEOPOL® W in comparison with the traditional technology applied in the production of automotive castings—RCS technology. For that purpose, four different core systems were used, including the inorganic binder GEOPOL® W20 with two different powder additives—GEOTEK W303 and GEOTEK W308B. Two commonly used resin coated sands, designated CB20B and DA30, were then selected as a comparison standard. With regard to the achieved results, the possibility of applying the innovative binder GEOPOL® W as a replacement for RCS can be positively confirmed. Partial results of individual experiments can be summarized in the following conclusions:

Both tested mixtures with inorganic binder showed high mechanical properties both hot and cold. The high mechanical properties of these molding mixtures are due to the high mobilization of the inorganic binder cured by dehydration processes in comparison with the usual chemical way (sodium silicate cured with $CO_2$-process, ester cured self-setting sodium silicate process).

The collapsibility of molding or core mixtures is one of the most important characteristics of a binder. The key factor is the residual strength, which is often referred to as the rate of collapsibility. This is especially in the case of inorganic binders in direct correlation with primary strengths according to the relationship—high primary strength of the binder system = high residual strength = poor collapsibility. From the obtained results it is evident that the resulting collapsibility of the mixtures with the studied inorganic binders was comparable or better in comparison with the RCS mixtures. In addition, according to the comparison of the results of measuring the transverse strengths of the tested mixtures, RCS mixtures (comparative standard) achieved more than 30% lower cold and hot strengths. Thus, with requirements for the same achieved strength, there was scope to reduce the dosage of GEOPOL® W20 binder, which should also have a positive effect on collapsibility.

From the obtained results it is evident that the difference in surface quality from cores in test castings was influenced by the different particle size distribution of the sands used, expressed by the value of AFS GN (or medium grain $d_{50}$). Even though it is a raw cast surface, the resulting surface roughness met the required quality—according to the values of Ra and Rz it corresponded to the requirements for surface roughness for castings produced by the HPDC method.

**Author Contributions:** Conceptualization, J.B., P.L. and K.J.; methodology, I.N. and M.D.; software, F.R.; validation, T.O., V.M. and M.F.; formal analysis, V.M. and J.B.; investigation, V.M., T.O. and J.B.; resources J.B.; data curation, I.K.; writing—original draft preparation J.B.; writing—review and editing K.J., I.K., V.M., M.F.; visualization, V.M. and J.B.; supervision, P.L.; project administration, V.M.; funding acquisition, P.L. and M.F. All authors have read and agreed to the published version of the manuscript.

**Funding:** This research was funded by the project No. CZ.02.1.01/0.0/0.0/17_049/0008399 from the EU and CR financial funds provided by the Operational Programme Research, Development and Education, Call 02_17_049 Long—Term Intersectoral Cooperation for ITI, Managing Authority: Czech Republic—Ministry of Education, Youth and Sports. This work was carried out in the support of projects of "Student Grant Competition" numbers SP2021/39 a SP2021/41.

**Institutional Review Board Statement:** Not applicable.

**Informed Consent Statement:** Not applicable.

**Data Availability Statement:** Not applicable.

**Conflicts of Interest:** The authors declare no conflict of interest.

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
