# Peer review of "Innovative Inorganic Binder Systems for the Production of Cores for Non-Ferrous Metal Alloys Reflecting the Product Quality Requirements"

_metals, doi:10.3390/met11050733_

Round 1

Reviewer 1 Report

This paper deals with a comparison of cores used in the foundry industry for non-ferrous castings. The comparison is between cores using inorganic binders against cores using organic binders. Properties such as strength, collapsibility, grain size, ignition loss, pH, conductivity, and the like were measured as well as casting properties such as surface roughness. The conclusion is that the innovative cores using inorganic binders behave better and they represent an improvement in cost and ecologically.

The paper is well organized and written and it is of interest to the foundry industry. Only a few issues must be addressed before publication:

  • Figure 3: The letters are too small.
  • Figure 4: include units in the Figure
  • Section 2 “materials and Methods”, line 173: The reference given to set the simulation (ref[14]) is not enough for the reader to understand the simulation. Please add the necessary text to explain the simulation's assumptions, equations, boundary, and initial conditions and explain in detail figures 5 and 6. By the way, letters are too small to see them (Figures 5 and 6).
  • Table3, Figure 7, Figure 9, Table 4: indicate if the test has an uncertainty level due to replicas.
  • Line 306: “…core stuck in the test casting. This value is the result of an evaluation on six cores (three castings),…”, use standard deviations in Figure 8.
  • Lines 332-333: …Ra and Rz are expressed as the average value from five measurements over the entire monitored 332 area of the test casting…; use standard deviations in Table 4.
  • Line 165: “…resp. impaired…”; use the complete word “resp.”

Author Response

Dear reviewer,

Thank you for your comments and suggestions to improve our manuscript. Based on your recommendations, we've edited the manuscript.

All changes are clearly highlighted in a different color in the revised manuscript.

Kind regards

Václav Merta

Reviewer 2 Report

Dear Authors,

The article is interesting and in general well-written. You described the experiments clearly and your conclusions are supported by the results. I did not find major issues to correct, however, a lot of minor issues should be addressed before the further processing:

  1. The title is too long and some parts are obvious for me. So. I suggest something like: “Innovative inorganic binder systems for the production of cores for non-ferrous alloys castings reflecting the product quality requirements”
  2. Only 3 email addresses are given what is against the journal’s rules.
  3. Lines 29-32: I think this sentences is misunderstanding. I think it would be better to replace word “connected” with “limited”.
  4. Formula (1) is too small to comfortable reading – please extend the fonts size.
  5. Line 58: I think “speed” could be replace with “rate” for better fit.
  6. Line 99: shell molding is first time referred here. Perhaps it is worth to mention earlier, including the Abstract?
  7. In Table 1 the set of process parameters for both tested sand was given. The explanations of their values were presented earlier. But what will happen when we use the same value of shooting pressure? This parameter seems to influence the mechanical properties much.
  8. Figure 3 must be significantly enlarged. Now it is completely unreadable.
  9. Line 165: “resp.” – please extend.
  10. I suggest to enlarge Figures 5 and 6.
  11. Line 242: I do not understand this: value of S the closer S is closer to 1
  12. Figures 11 and 12 are the same.

After you consider my remarks and correct the mistakes I do recommend the article for publishing.

Sincerely,

Reviewer

Author Response

(The authors gave the same response as above.)
